# Influence of Attachment Anxiety on the Relationship between Loneliness and Depression among Long-Term Care Residents

**DOI:** 10.3390/healthcare9121675

**Published:** 2021-12-03

**Authors:** Suthikarn Arunrasameesopa, Nahathai Wongpakaran, Tinakon Wongpakaran

**Affiliations:** Department of Psychiatry, Faculty of Medicine, Chiang Mai University, Chiang Mai 50200, Thailand; suthikarn.a@cmu.ac.th (S.A.); tinakon.w@cmu.ac.th (T.W.)

**Keywords:** depressive, lonely, isolation, COVID-19, nursing home, old age, attachment style

## Abstract

Background: Attachment styles influence an individual’s ability to maintain relationships throughout their lifespan. Insecure attachment has been associated with many aspects of mental health. The study aimed to explore the influence of attachment on loneliness and depression among old-age residents in long-term care facilities. Methods: A survey was conducted among residents from long term care facilities (LTC) in Thailand during the COVID-19 pandemic; 221 older people participated in the study. The mean aged was 73.60 ± 7.45 years old, and 57.5% female. All participants completed the Geriatric Depression Scale, the six-item revised version of the University of California Los Angeles Loneliness Scale, and the Revised Experience of Close Relationships questionnaire to measure depressive symptoms, loneliness, and attachment style. Hierarchical multiple regression analysis was performed to find the predictors for depressive symptoms. Results: In all, 40.7% of residents in LTC facilities experienced depression. Male sex, education, loneliness, and attachment anxiety predicted the increased level of depressive symptoms. Conclusion: Loneliness was confirmed to be significantly associated with depressive symptoms, and attachment anxiety was found to be a strong predicting factor of depressive symptoms. Intervention and research regarding reducing loneliness and attachment anxiety should be promoted in LTC settings.

## 1. Introduction

It has been estimated that about 700 million people worldwide are older people, and that number is estimated to double in 2050 [1]. Thailand is one of the countries in transition to Aged Society, and the population of older people is predicted to reach 30% of the total population by 2050 [2]. Changing the family structure from extended to nuclear family and having fewer children in Thai families [3] are creating challenges in caregiving for older adults [4]. The aging phenomenon can also be explained by “second demographic transition” [5]. Some have to stay in long-term care (LTC) facilities. The World Health Organization has recommended that to deal with this transition, key issues for public health actions are using the concept of a person-centered approach applied in long-term care (LTC) facilities, helping older people by reducing inequal opportunities to access to health care services, and ensuring autonomy, dignity, integrity, freedom and independence [6].

A systematic review study revealed that the prevalence of depression was 5 to 25% among residents in LTC homes [7]; in Thailand, it was found to be up to 23% [8]. Many factors have been found to be associated with depression among residents in LTC settings [9,10,11,12,13], including psychological distress or fear related to COVID-19 during this pandemic [14]. Loneliness is one of the common factors experienced among older people, and associated with depression [13] and a lower quality of life of residents in LTC settings [15,16]. Loneliness is the subjective feeling of lacking intimate and close relationships to others [17]. Different measurements were developed according to different understandings about loneliness; The De Jong Gierveld Loneliness Scale (DJGLS) considers loneliness from three perspectives (types of loneliness, the perspective of time, and emotional aspects) [17]; the UCLA loneliness scale contains two dimensions, i.e., negatively and positively worded items [18]. Studies have shown that 11.6 to 85.3% of people living in LTC settings have experienced loneliness [15,19]. Factors associated with loneliness include living alone, being unsatisfied with life, being at risk for undernutrition, having mental problems, having a weak sense of coherence, and not having contact with neighbors [19]. For residents in LTC settings, investigators found that a lack of self-determination due to being institutionalized and having unfulfilled needs for meaningful relationships played crucial roles in loneliness [20]. In addition to the association with depression, loneliness could be a risk factor for subsequent cognitive impairment [21].

Despite the fact that loneliness is related to many factors and many interventions have been adopted to mitigate such feeling especially during the COVID-19 pandemic [22], one psychological construct related to loneliness that might not have been explored sufficiently is individual attachment style. Attachment has been associated with many aspects of mental health, especially loneliness and depression. Insecure attachment appears to be associated with depressive symptoms and loneliness among adults [23,24,25,26,27]. Attachment was also found to be related to suicidality in that attachment patterns predicted medical lethality, amplifying the levels of mental pain [28]. Attachment theory is mainly influenced by the work of John Bowlby and Mary Ainsworth, focusing on the relationship between infants and their mothers, then considering the consequence of their relationship and the long-term relationship to others throughout their lifespan [29]. Secure attachment brought people a sense of self-worth and self-confidence, and they were more likely to enjoy stable relationships [30]. On the contrary, insecure attachment style created low self-esteem, unstable relationships, difficulties regulating emotions, increased risk of feelings of loneliness or developing depression and psychological distress [31]. Attachment style is classified by secure attachment, fearful insecure attachment, preoccupied insecure attachment, and dismissive insecure attachment [32]. Few studies have been conducted regarding the role of attachment on psychopathology among older people. One study confirmed its importance, similar to what was found among younger people in that those higher levels of attachment anxiety significantly increased the suicide risk among older adults [33]. However, research in special settings such as LTC homes remains scarce. It would be interesting to learn how the existing attachment style of older adults residing in this setting affect easily aroused feelings of loneliness, influencing the development of depression. Our study aimed to explore depression, feelings of loneliness, and attachment style among older people in LTC settings in Thailand, and also the predictive ability of these independent variables on depression. We hypothesized, based on the developmentally theoretical underpinning and related research evidence, that both loneliness, attachment anxiety and attachment avoidance would significantly relate to depression.

## 2. Subjects and Methods

This constituted a cross-sectional observational study conducted in LTC settings in Thailand from December 2020 to July 2021. A single population proportion formula with an assumption of 95% confidence level, the expected proportion with 7% absolute precision (margin of error), 23% previous prevalence of depression in LTC settings [8] and adjusted for finite population of 1292 and a nonacceptance rate of 15% were used to determine a minimum sample size of 145.

### 2.1. Participants

We enrolled residents aged over 60 years in LTC homes who were able to communicate orally in Thai. The total number of residents in Thailand in 2021 was 1292 in 12 government’s LTC homes all over Thailand [34]. The inclusion criteria included LTC residents who were: (1) aged 60 years old or higher, (2) able to understand and communicate in Thai, and (3) able to complete questionnaires on their own. We excluded residents who had: (1) weakness in the dominant hand, hearing loss or visual loss to a degree that interfered communication, (2) received a diagnosis of dementia, and/or (3) obtained a Mini-Cog score of less than 3 points [35,36]. Of 294 residents recruited, 247 participants were enrolled, 11 participants met the exclusion criteria, and 15 participants had incomplete data. Finally, we obtained 221 participants for this analysis (Figure 1). All the participants were asked to complete the questionnaires. Informed consent was obtained from all participants from the beginning before proceeding to further steps. This research was approved by the Ethics Committee of the Faculty of Medicine, Chiang Mai University, Thailand.

### 2.2. Instruments

The 18-item revised Experience of Close Relationships Questionnaire (ECR-R-18): This survey comprises a self-rating measurement assessing adult attachment style, yielding the scores of attachment anxiety and attachment avoidance [37]. It consists of 18 items: 9 questions for attachment anxiety and 9 questions for avoidance. Each question employs a 7-level Likert scale, where 1 is fully disagree, and 7 is fully agree. Higher scores indicate a higher level of anxiety or avoidance, and the scale is valid and reliable [38,39]. In this study, the Cronbach’s alpha was 0.90 and 0.82 for anxiety and avoidance subscale, respectively.

The 6-item geriatric depression scale (GDS-6): This screening instrument for depression among older adults consists of 6 dichotomous-type (“Yes/No”) questions to evaluate depressive symptoms the score ranges from 0 to 6 points. Higher scores indicate higher levels of depression [40]. The GDS cut-off score ≥ 2, suggests depression [40] In this study, the Cronbach’s alpha was 0.70.

The Revised University of California Los Angeles Loneliness scale (RULS-6): This tool evaluates subjective feelings of loneliness and social isolation. It consists of 6 questions, with a score of 1 to 4 in each question, with 1 being never and 4 being often. Higher scores indicate higher levels of feelings of loneliness [41]. In this study, the Cronbach’s alpha was 0.79.

Mini-Cog: This tool is a brief cognitive screening test, which combines two parts i.e., a short-term memory test (the recall of 3 unrelated words–total 3 points) and the clock drawing test (total 2 points). The total scores range from 0 to 5. Scores of 2 and lower indicate cognitive impairment [35,36]

### 2.3. Statistical Analysis

Descriptive statistics, e.g., frequency, percentage, mean and standard deviation, were used for demographic data, e.g., age, sex, years of education, marital status and measurement scores, i.e., ECR-R-18, GDS-6, and RULS-6. Pearson correlations were analyzed to correlate between measurement scores. Pair variables significantly correlating with *p* < 0.05 were used for multiple regression. Hierarchical multiple regression was analyzed to demonstrate how each predictor affected the depressive outcome when holding all other predictors constant. In this case, we endeavored to see how the impact of loneliness on depression would change when attachment was considered in the models. We began with demographic data, i.e., sex, age, and education, as predictors in the first model, followed by the RULS score in the second model. Finally, ECR-R anxiety scores and ECR-R avoidance scores were added in the third model. The R square change was used to demonstrate the effect of the new set of predictors when they were added into the model. The R square change, significantly greater than zero, was inferred to have incremental predictive power. The R square change (increment) from Model 1 to Model 2 was computed as ΔR^2^ = Model 2 R^2^-Model1 R^2^. The same rule was applied to Model 3. Statistical significance was set at <0.05 and all analyses were performed using IBM SPSS, Version 22.

## 3. Results

In total, 221 participants completed the questionnaire. A total of 127 participants were female (57.5%) with a mean age of 73.60 years (SD = 7.45). Most participants’ marital status was single (44.8%). The mean score of attachment-anxiety was 3.92 and of attachment-avoidance was 3.39. The mean GDS score was 2.71 (SD = 1.22). Depression based on GDS cut-off were found among 90 participants (40.7%). The demographic and clinical data are shown in Table 1.

Table 2 shows the zero-order correlation between variables. Male sex was positively related to GDS scores (r = −0.22, *p* < 0.01), whereas education was negatively associated with GDS scores. As expected, GDS was positively associated with RULS (*p* < 0.001). ECR-R anxiety but not ECR-R avoidance was associated with GDS, RULS, and age (*p* < 0.01)

We analyzed hierarchical multiple regression to predict depression by attachment avoidance, attachment anxiety and loneliness. Demographic variables were entered as covariates on Step 1. Male sex was a significantly positive predictor for depression, whereas education was a negative predictor in Model 1 (β = −0.19, *p* <0.01; β = −0.19, *p* < 0.01, respectively). RULS scores were added in a second step, and significantly predicted the GDS score (β = 0.33, *p* < 0.001). R2 significantly increased in Model 2 compared with Model 1 (0.11, *p* < 0.001). ECRR-anxiety scores were added in the third step and found to be a significant additional predictor of depression (β = 0.25, *p* < 0.001). R2 significantly increased in Model 3 compared with Model 2 (0.05, *p* < 0.001). Lastly, we added ECRR-avoidance, but did not predict the GDS score (β = 0.11, *p* ≥ 0.05). R2 did not significantly change in Model 4 from Model 3 (0.009, *p* = 0.109). The R2 increased from 7% in the Model 1 to 23% in the final model (Table 3)

## 4. Discussion

This study aimed to explore the prevalence of depression and examine how depression was influenced by loneliness, and attachment among older adults residing in LTC homes. Depression was found in 40.7% of residents in LTC homes, which was relatively high. Related studies have revealed a wide range in the prevalence of depression among residents in LTC homes, 5 and 82%, and usually higher than in the community [7,42,43,44,45,46,47]. Compared with the report of the prevalence of 23% of depression in LTC homes in our previous study, and 23.7% in a tertiary care setting [8,48], the present result was strikingly high during the COVID-19 pandemic. This supports the prior research [49].

In contrast to many studies, we have found that males experienced higher depression than females. Providing an easy explanation for this contradicting result would be difficult compared with the prior evidence that females were more likely to experience depression. One assumption has to do with the COVID-19 situation. Quite possibly, the stressful event of social isolation and any measure used regarding COVID-19 prevention could psychologically impact males more than females in LTC homes [50,51]. However, undetermined factors might be involved, needing further study.

As expected, subjects with low education levels were more likely to develop depression, but this was not the case for age, as found in other studies [43,52,53,54,55,56]. This could be because most participants were less old, and the range of age distribution was narrow compared to other related studies. In terms of loneliness, the relationship between loneliness and depression in LTC homes remained during the COVID-19 pandemic [57,58,59]. During the COVID-19 pandemic, the depression rate increased as social isolation would aggravate the feelings of loneliness, rendering the eventual depression outcome [49,60].

The relationship between loneliness and depression was, however, lessened when attachment was added to the model. Our findings indicated that only a high level of attachment anxiety, but not attachment avoidance, predicted the increase in depression. Attachment anxiety represents a fearful style or preoccupied style and explained the increased variance of depression from 18 to 23%. The fact that attachment anxiety (fearful or preoccupied style) but not attachment avoidance (dismissing style) was associated with depression was consistent with some related studies [25,61]. Some investigators found that fearful attachment and preoccupied attachment were associated with depression through neediness. A dismissive style was unrelated to neediness; hence, it did not predict depression [62]. Moreover, attachment anxiety may have been linked with depression through low self-esteem [63], and also through interpersonal sensitivity [64].

Various interventions, especially psychological therapies and leisure/skill development interventions, are provided to help reduce loneliness [65]. Some researchers suggest active coping strategies with culturally based or religious reliance, and social engagement with peers can be incorporated in healthcare services and social programs to assist these older adults [66]. Moreover, psychosocial interventions such as reminiscence therapy, support groups, or videoconferences are positive ways to create a social connection to improve loneliness and depression among residents in LTC homes [67].

The present results suggested that methods or strategies should be combined with the approach for attachment insecurity. Basically, individuals with preoccupied attachment usually express their desire to feel wanted. They may require frequent reassurance and approval. That can create burdens on others or caregivers and make it difficult to maintain. People with fearful attachment are even more difficult to deal with. They still want relationships and may actively search for them. However, when the relationship becomes too committed, the person with a fearful avoidant attachment may respond by simply running away from the relationship [68]. Therefore, understanding and helping individuals to cope with stress and loneliness based on the understanding of the attachment concept would be important [69]. For example, staff and caregivers in LTC homes may be aware of the importance of constant reassurance or appropriate approaches while providing intervention for combatting loneliness. However, intervention incorporated with the attachment concept specifically designed to deal with loneliness should experiment in further research.

## 5. Limitations

This study constitutes one of the first to study attachment style associated with loneliness and depression in LTC homes. Some limitations to be addressed are discussed below. We collected information from participants during the COVID-19 pandemic, making it difficult to process and collect data from participants because preventive measures in COVID-19, such as maintaining a physical distance, were applied, resulting in time limitations in this process. Second, we established inclusion criteria that specified only people able to communicate orally in Thai and having intact hearing or vision.

## 6. Conclusions

Nearly one half of residents in LTC facilities experienced depression, which was related to loneliness. Attachment style, developed during the early years of life, remained a critical psychological process among older adults. Attachment anxiety was found to be the strongest predictor among all independent variables in predicting depressive symptoms. The application of the results to provide intervention for older adults with depressive feelings only based on their anxious attachments was discussed.

Replication studies in different cultures are encouraged. Finding out whether the COVID-19 pandemic has an influence on the relation among these variables may be of benefit. Further study regarding intervention to reduce loneliness among LTC home residents may also be warranted.

## Figures and Tables

**Figure 1 healthcare-09-01675-f001:**
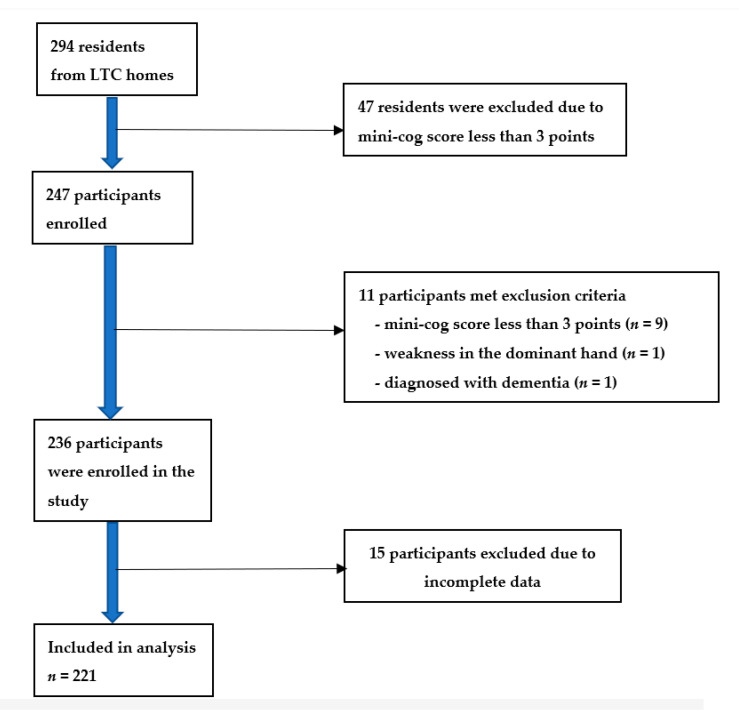
Flowchart of the study.

**Table 1 healthcare-09-01675-t001:** Participants’ characteristics (*N* = 221).

Variable	*N* (%)/Mean ± SD
Socio-demographic	
Age, Mean (SD)	73.60 ± 7.45
Sex, *N* (% female)	127(57.5)
Education (years),Mean (SD)	6.87 ± 4.53
Marital status, *N* (%)	
Single	99 (44.8)
Married	11 (5)
Divorced	27 (12.2)
Widowed	83(37.6)
No data	1(0.4)
Occupation, *N* (%)	
Employee	85 (38.5)
Government officer	19 (8.6)
Personal business	31 (14)
Unemployed	76 (34.4)
Other	10 (4.5)
Clinical data	
ECR-R anxiety, Mean ± SD	3.92 ± 1.43
ECR-R avoidance, Mean ± SD	3.39 ± 1.22
GDS total, Mean ± SD	2.71 ± 1.22
Depression based on GDS cut-off, *N* (%)	90(40.7)
RULS, Mean ± SD	13.67 ± 4.19

Note: ECR-R, 18-item revised Experience of Close Relationships Questionnaire; GDS, the geriatric depression scale; RULS, the Revised University of California Los Angeles Loneliness Scale.

**Table 2 healthcare-09-01675-t002:** Correlation matrix.

Variable	1	2	3	4	5	6	7
1.Sex	1	0.28 ***	0.02	0.03	−0.22 **	−0.05	−0.13
2.Age		1	0.05	0.11	−0.11	0.02	−0.19 **
3.Education			1	−0.04	−0.17 *	−0.01	−0.12
4.RULS				1	0.39 ***	0.06	0.20 **
5.GDS					1	−0.00	0.35 ***
6.ECRR-avoidance						1	−0.44 ***
7.ECRR-anxiety							1

* *p* < 0.05; ** *p* < 0.01; *** *p* < 0.001, Note: ECR-R, 18-item revised Experience of Close Relationships Questionnaire; GDS, the geriatric depression scale; RULS, the Revised University of California Los Angeles Loneliness Scale.

**Table 3 healthcare-09-01675-t003:** Hierarchical regression of attachment and loneliness in depression.

Predictor	Model 1			Model 2			Model 3			Model 4		
	**B**	**β**	**SE**	**B**	**β**	**SE**	**B**	**β**	**SE**	**B**	**β**	**SE**
Sex	−0.56	−0.19 **	0.20	−0.56	−0.19 **	0.19	−0.50	−0.17 **	0.18	−0.47	−0.16 *	0.18
Age (years)	−0.01	−0.05	0.01	−0.02	−0.09	0.01	−0.01	−0.04	0.01	−0.01	−0.04	0.01
Education (years)	−0.06	−0.19 **	0.02	−0.06	−0.17 **	0.02	−0.05	−0.15 *	0.02	−0.05	−0.14 *	0.02
RULS				0.69	0.33 ***	0.13	0.58	0.28 ***	0.13	0.54	0.26 ***	0.13
ECRR-anxiety							0.25	0.25 ***	0.06	0.31	0.30 ***	0.07
ECRR-avoidance										0.13	0.11	0.08
R^2^	0.07			0.18			0.23			0.23		
R^2^ change	0.08			0.11			0.05			0.009		

Model 1 included demographic variables, Model 2 included loneliness, Model 3 included demographic, loneliness, and attachment anxiety, and Model 4 included demographic, loneliness, attachment anxiety, and attachment avoidance; sex: male = 1, female = 2, * *p* < 0.05; ** *p* < 0.01; *** *p* < 0.001, B = unstandardized regression coefficient, β = standardized regression coefficient, SE = standard error.

## Data Availability

The datasets used and/or analyzed during the current study are available from the corresponding author upon reasonable request.

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
