# Peer review of "Influence of Attachment Anxiety on the Relationship between Loneliness and Depression among Long-Term Care Residents"

_healthcare, 2021, doi:10.3390/healthcare9121675_

Round 1

Reviewer 1 Report

I consider the following comments to improve the paper

1.-  In the introduction section, in the phrase: "Thai families [2]. are creating", please eliminate the point "."

2.- In this paragraph of introduction: "attachment style is characterized by secure attachment", I would change the word "characterized" by "classified"  

3.- In the section of "material and methods": please include the reference of "mini-cog", in relation with this test: does the authors refer to mini-mental or is another study?

4.- In the section of "material and methods": I would change the word "material" by "subjects"

5.- In table 1 of results, the percentage of "marital status" sums 99.6% instead 100%, and the sum of the studied individuals is 220 instead 221, please correct. 

6.- In the results section, the paragraph: "ECR-R was associated with anxiety, but avoidance was associated with GDS, RULS, and age (p<0.01)", does not reflects in the Table 2, where the associated variable with GDS, RULS and age is the "ECRR-anxiety" but not "ECRR-avoidance". Please correct the writing. 

7.- In the results section in the paragraph: "sex and education years were negatively significant predictors for depression", I suggest to clear that the male sex was positively associated with depression. 

8.- Instead the "hierarchical regression analysis" I suggest to perform a multiple regression analysis, without the steps, and use the "stepwise" method", this test permits to construct models where all the included variables are significant, because in the presented models many variables are not significant: by example, in model 4, the variables ECRR-avoidance and age, are not significant. Remember that for a model to be valid needs to have all the included variables significantly associated with the dependent variable. 

9.- In the discussion section: in the phrase: "the fact that attachment anxiety...,not attachment avoidance..." is not clear, I suggest to add "but" before "not attachment". 

10.- The writing of the discussion section could be improving by diminishing the short phrases separated by points, and to add commas "," or point and commas ";" instead.   

11.- Finally, in the abstract section, the phrase: "and attachment anxiety appeared to provide a mediating effect on the relationship between loneliness and depressive symptoms." , the adjective of "mediating effect" gives an impression of having performed a path analysis (also named: structural equation modeling), and these kind of multivariate analysis was not performed in this study. I suggest to eliminate this phrase, because there is not a clear evidence that attachment anxiety is an intermediate or mediating variable according to the results given.  

Author Response

1.-  In the introduction section, in the phrase: "Thai families [2]. are creating", please eliminate the point "."

Response: Thank you for pointing out our mistake. We have removed it.

2.- In this paragraph of introduction: "attachment style is characterized by secure attachment", I would change the word "characterized" by "classified"  

Response: We agree with the comment and have replaced the word as suggested. Please see the blue text in page 2.

3.- In the section of "material and methods": please include the reference of "mini-cog", in relation with this test: does the authors refer to mini-mental or is another study?

Response: We apologize for the missing citation. The Mini-Cog we used is not the same as the MMSE, but the Mini-Cog by Borson et al., 2003. Please see our revision and added citations on lines 103-104.

We have also added the Mini-Cog and its details in the Instruments section as read, ‘2.2.3 Mini-Cog: This tool is a brief cognitive screening test, which combines two parts, i.e., a short-term memory test (the recall of 3 unrelated words – total 3 points) and the clock drawing test (total 2 points). The total scores range from 0 to 5. Scores of 2 and lower indicates cognitive impairment [37, 38]. Please see page 4.

4.- In the section of "material and methods": I would change the word "material" by "subjects"

Response: We have replaced the word as suggested. Please see page 2.

5.- In table 1 of results, the percentage of "marital status" sums 99.6% instead 100%, and the sum of the studied individuals is 220 instead 221, please correct. 

Response: Thank you for pointing out this mistake. There was one missing data regarding marital status as one participant did not reveal the information. We have corrected the information. Please see blue texts in the Table 1 as suggested.

6.- In the results section, the paragraph: "ECR-R was associated with anxiety, but avoidance was associated with GDS, RULS, and age (p<0.01)", does not reflects in the Table 2, where the associated variable with GDS, RULS and age is the "ECRR-anxiety" but not "ECRR-avoidance". Please correct the writing. 

Response: Thank you for pointing out our mistake. We have rewritten this text to; “ECR-R anxiety but not ECR-R avoidance was associated with GDS, RULS, and age (p<.01)”. Please see page 5.

7.- In the results section in the paragraph: "sex and education years were negatively significant predictors for depression", I suggest to clear that the male sex was positively associated with depression. 

Response: Thanks. We agree and have revised this text by replacing it with “Male sex was significantly a positive predictor for depression, whereas education was a negative predictor...”. Pls see page 6.

8.- Instead the "hierarchical regression analysis" I suggest to perform a multiple regression analysis, without the steps, and use the "stepwise" method", this test permits to construct models where all the included variables are significant, because in the presented models many variables are not significant: by example, in model 4, the variables ECRR-avoidance and age, are not significant. Remember that for a model to be valid needs to have all the included variables significantly associated with the dependent variable. 

Response: Thank you for your suggestion. We agree that both hierarchical regression and multiple regression methods could make the same results. The reasons that we used the hierarchical multiple regression was driven from theoretical rather than statistical reason. ECRR-avoidance, despite not correlated with RULS and GDS, was still needed in the model to determine the attachment style (please see Line 216-218 in our discussion and Reference no. 26 and 64). In addition, we would like to show how RULS changed after ECRR was added, and how ECRR-anxiety was influenced by ECRR-avoidance when it was added. Despite the fact that ECRR-avoidance was not significantly correlated with RULS and GDS), it increased the effect size of ECRR-anxiety when ECRR-avoidance was added because they were correlated. Also, the step by step process using hierarchical regression could demonstrate to what extent the effect size of each variable on the outcomes was changed by another variable in mediating fashion. With this advantage of hierarchical regression over stepwise method, which allows the computer to decide which variable to enter at what stage, we, hence, kept the hierarchical multiple regression.

let the computer decide which terms to enter at what stage

9.- In the discussion section: in the phrase: "the fact that attachment anxiety...,not attachment avoidance..." is not clear, I suggest to add "but" before "not attachment".

Response: Thanks. We added “but” as suggested. Please see page 7. 

10.- The writing of the discussion section could be improving by diminishing the short phrases separated by points, and to add commas "," or point and commas ";" instead.   

Response: Thank you for your suggestion, we have this manuscript seen by an English native speaker.

11.- Finally, in the abstract section, the phrase: "and attachment anxiety appeared to provide a mediating effect on the relationship between loneliness and depressive symptoms." , the adjective of "mediating effect" gives an impression of having performed a path analysis (also named: structural equation modeling), and these kind of multivariate analysis was not performed in this study. I suggest to eliminate this phrase, because there is not a clear evidence that attachment anxiety is an intermediate or mediating variable according to the results given. 

Response: Thank you for your comment. We have changed this phrase to “and attachment anxiety was found to be a strong predicting factor of depressive symptoms”

Reviewer 2 Report

Comments and Suggestions for Authors

The manuscript Influence of attachment anxiety on the relationship between loneliness and depression among long term care residents aimed to explore the influence of attachment on loneliness and depression among old age residents in long term care facilities.

In my opinion, the theme is relevant, although I would like to point out some aspects that could be improved. Some of them require a major modification of the paper:

Title

The title should include the geographical context of the study to avoid extrapolating to the generality that which is particular. It is also convenient to include that it takes place during the COVID-19 pandemic, as this could bias the study.

Abstract

  1. It´s great in my opinión

Introduction

  • The authors point out that it is estimated that about 2.1 billion people worldwide are elderly. This number does not coincide with that of other research, so please indicate the reference.

  • It is recommended to speak of the concept of a "second demographic transition" to explain the ageing phenomenon. It may be useful in this respect:

-VAN DE KAA, D. J. (2002) «The idea of a second demographic transition in industrialized countries» en Sixth Welfare Policy Seminar of the National Institute of Population and Social Security, Tokyo.

  • It is recommended to discuss WHO's position on the public health problem of ageing. It may be useful in this regard:

-World Health Organization. World report on ageing and health. World Health Organization, 2015.

  • The concept of loneliness is complex, and is not sufficiently addressed. It is not very clear what kind of loneliness the authors are referring to. It would be highly advisable to elaborate on this point. The relationship they establish is not the same if they refer to one type of loneliness or another. Some notes are given in papers such as:

Rivero Jiménez, Borja, David Conde-Caballero, and Lorenzo Mariano Juárez. "Loneliness Among the Elderly in Rural Contexts: A Mixed-Method Study Protocol." International Journal of Qualitative Methods 20 (2021): 1609406921996861.

  • In my opinion, the introduction could be improve for all these reasons.

Methods

  • In my opinion, it is sufficient. The sample is sufficient and the method seems to me to be adequate.

Results

  • The authors say that the situation caused by COVID is important, and therefore I suggest you include it in the title as a determinant. What is not clear to me is why the consequences are different for men and women. I think this point deserves further explanation.

Discussion

  • A discussion section with other research in the same vein is missing. This is essential in order to be accepted for publication. In my opinion this is the main problem with the paper.

Conclusions

  • The conclusions are very short and need to be expanded. It is good that possible future lines of research are indicated.

Author Response

Introduction

  • The authors point out that it is estimated that about 2.1 billion people worldwide are elderly. This number does not coincide with that of other research, so please indicate the reference.

 Response: Thank you for your comment. We rechecked the data and citation and have revised them. Please find the text, “It has been estimated that about 700 million people worldwide are older people, and that number is estimated to double in 2050 [1]. Thailand is one of the countries in transition to Aged Society, and the population of older people are predicted to reach 30% of the total population by 2050 [2]” in the Introduction section.

  • It is recommended to speak of the concept of a "second demographic transition" to explain the ageing phenomenon. It may be useful in this respect:

-VAN DE KAA, D. J. (2002) «The idea of a second demographic transition in industrialized countries» en Sixth Welfare Policy Seminar of the National Institute of Population and Social Security, Tokyo.

Response: Thank you for your suggestion. We have added the text, ‘The aging phenomenon can also be explained by ‘second demographic transition [5] in the Introduction section.

  • It is recommended to discuss WHO's position on the public health problem of ageing. It may be useful in this regard:

-World Health Organization. World report on ageing and health. World Health Organization, 2015.

 Response: Thank you for your suggestion. We have added the text, “World Health Organization has recommended that to deal with this transition, key issues for public health actions are using the concept of person-centered applied in long term care (LTC) facilities, helping older people reducing inequal opportunities to access to health care services, and ensuring autonomy, dignity, integrity, freedom and independence”, in the Introduction.

  • The concept of loneliness is complex, and is not sufficiently addressed. It is not very clear what kind of loneliness the authors are referring to. It would be highly advisable to elaborate on this point. The relationship they establish is not the same if they refer to one type of loneliness or another. Some notes are given in papers such as:
  • Rivero Jiménez, Borja, David Conde-Caballero, and Lorenzo Mariano Juárez. "Loneliness Among the Elderly in Rural Contexts: A Mixed-Method Study Protocol." International Journal of Qualitative Methods 20 (2021): 1609406921996861.

Response: Thank you for your advice. We have added this content, “Loneliness is a subjective feeling of lacking intimate and close relationships to others [17]. Different measurements were developed according to different understanding about loneliness; The De Jong Gierveld Loneliness Scale (DJGLS) considers loneliness from three perspectives (types of loneliness, the perspective of time, and emotional aspects) [18]; the UCLA loneliness scale contains two dimensions, i.e., negatively and positively worded items [19], etc.” in the introduction section. Please see page 2.

Methods

  • In my opinion, it is sufficient. The sample is sufficient and the method seems to me to be adequate.

Response: Thank you.

Results

  • The authors say that the situation caused by COVID is important, and therefore I suggest you include it in the title as a determinant. What is not clear to me is why the consequences are different for men and women. I think this point deserves further explanation.

Response: We apologize for not including COVID-19 in the title because we did not have the information before the COVID-19 spread. We are unable to demonstrate that the finding was determined by the COVID-19 pandemic either. The fact that depressive symptoms were higher among males than females in this study is difficult to explain. It’s not clear to us either. Qualitative or mixed method designs may be required for further study.

Discussion

  • A discussion section with other research in the same vein is missing. This is essential in order to be accepted for publication. In my opinion this is the main problem with the paper.

Response: Thank you. We have conducted a vigorous literature search regarding relationship among attachment, depression, and loneliness in long term care facilities and found no publication. To the best of our knowledge, this study is the first study to demonstrate these findings.

Conclusions

  • The conclusions are very short and need to be expanded. It is good that possible future lines of research are indicated.

Response: We have added the text, “Replication studies in different cultures are encouraged. Finding out whether the COVID-19 pandemic has an influence on the relations among these variables may be of benefit. Further study regarding intervention study to reduce loneliness among LTC home residents would also be warranted.” to the Conclusion section.

Round 2

Reviewer 1 Report

I observed that the manuscript improved significantly and is ready for publication.     

Author Response

Thank you very much./Nahathai

Reviewer 2 Report

Thank you very much and congratulations for the work done. 

Author Response

Thank you very much./Nahathai